# Closing the gap towards super-long suspension bridges using computational morphogenesis

Mads Baandrup [1,2✉], Ole Sigmund [3], Henrik Polk[2] & Niels Aage [3✉]

Girder design for suspension bridges has remained largely unchanged for the past 60 years. However, for future super-long bridges, aiming at record-breaking spans beyond 3 km, the girder weight is a limiting factor. Here we report on a design concept, inspired by computational morphogenesis procedures, demonstrating possible weight savings in excess of 28 percent while maintaining manufacturability. Although morphogenesis procedures are rarely used in civil engineering, often due to complicated designs, we demonstrate that even a crude extraction of the main features of the optimized design, followed by a simple parametric optimization, results in hitherto unseen weight reductions. We expect that further studies of the proposed design, as well as applications to other structures, will lead to even greater weight savings and reductions in carbon footprint in a construction industry, currently responsible for 39 percent of the world's $CO_2$ emissions.

[1] Department of Civil Engineering, Technical University of Denmark, Brovej, Building 118, 2800 Kongens Lyngby, Denmark. [2] Department of Major Bridges International, COWI A/S, Parallelvej 2, 2800 Kongens Lyngby, Denmark. [3] Department of Mechanical Engineering, Technical University of Denmark, Nils Koppels Allé, Building 404, 2800 Kongens Lyngby, Denmark. ✉email: mjbp@cowi.com; naage@mek.dtu.dk

Since the opening of the Union Bridge on the Scotland–England border in 1820, suspension bridges have played a key role in civil infrastructure, facilitating fixed links on sites formerly separated by large distances or deep waters, and have become world-known landmarks such as the Brooklyn and Golden Gate Bridges. With the longest bridge spans doubling approximately every 50 years, soon passing 2 km, and with eight out of the ten longest spans in history constructed within the past 15 years, the evolution of bridges has been significant. However, since the 1950s, the conventional design concept for bridge girders, the orthotropic closed steel box-girder (Fig. 1 bottom left), has remained largely unchanged[1–3], despite significant design challenges due to inherent fatigue issues[4,5] (the phenomenon of failure from repeated cyclic loading). The simple and manufacturing friendly plate-based concept consists of outer skin plates stiffened longitudinally by troughs and transversely by diaphragms every 4–5 m, built with either trusses or solid plates ("conventional design" shown in Fig. 2 and indicated by blue in Fig. 1). Consequently, loads are carried indirectly in a non-natural zig-zag fashion, from bridge deck to hangers, causing stress concentrations and eventually, fatigue problems.

Soon, with plans for bridges spanning beyond 3 km in Norway, Italy, and Indonesia, and approaching the theoretical limit[6] of 5 km over the Strait of Gibraltar, self-weight will by far, be the governing load, exceeding 90% of the total bridge loads (including traffic, wind, and seismic). Furthermore, considering that the construction industry accounts for 39% of the world's $CO_2$ emissions[7], attention must be shifted from construction cost to reducing material consumption and hence, self-weight. With weak hopes for new mass-manufacturable, light-weight-high-strength construction materials, and keeping the fatigue problems in mind, the focus must be turned to the identification of new and material-efficient bridge designs.

Previously, research focus has concentrated on reducing fatigue problems[8–10], while very little attention has been given on identifying new design concepts[11], where no works focus particularly on weight or $CO_2$ emissions reductions. Furthermore, a recent gradient-based parametric optimization study showed very little room for improvement without altering the geometrical concept[12]. Hence, to pave the way for super-long suspension bridges and to reduce environmental impact, new and innovative design concepts are needed.

We seek to reinvent the more than 60-year-old girder design concept. With a focus on redesigning and decreasing the girder weight, knock-on effects will ensure similar weight savings in the remaining bridge structure (main cables, towers, foundations, and anchor blocks)[6]. Our study illustrates the benefits of applying a morphogenesis procedure with unrestricted design freedom in the early phase of civil engineering design development, and how a simplistic interpretation of the highly detailed optimized geometry, results in a simple, cost-efficient and constructible design concept. It is expected that similar conclusions can be drawn for applications to other structural problems, particularly within civil engineering.

## Results

**Topology optimization of bridge girder.** To allow for greatest possible design freedom, a giga-scale computational morphogenesis procedure[13,14] is used to identify the new design concept. The original concept, dating back to 1988[15], known as topology optimization, identifies structural solutions with unrestricted design freedom. The iterative numerical procedure redistributes material within a predetermined design domain to optimize a set of performance targets for a given set of loads and boundary conditions[16]. During the past few decades, the approach has become the preferred design tool in automotive[17] and aerospace[18] industries, but less so in civil engineering[19], though with a few studies of high-rise buildings[20], concrete reinforcement[21], as well as bridges[22–24]. The limited number of applications in civil

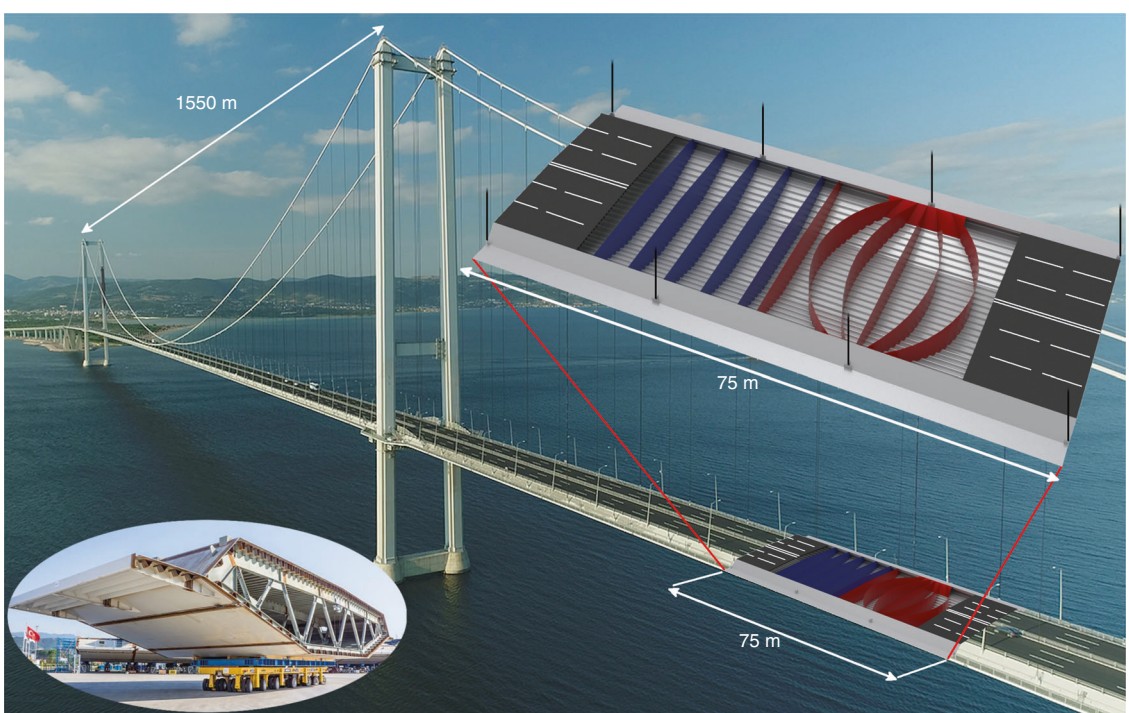

**Fig. 1 Conventional and interpreted girder design.** In the background, the Osman Gazi Bridge is shown with an indication of the main span of 1550 m. Insert in the lower left shows a conventional girder section of the Osman Gazi Bridge with truss diaphragms. In the upper right insert, three sections of the continuous girder are shown with solid diaphragms. The conventional design (blue) and the interpreted design (red), extracted from the optimization result, are shown in each of the two center sections, respectively. Here, the fixed top layer is removed to reveal the internal details.

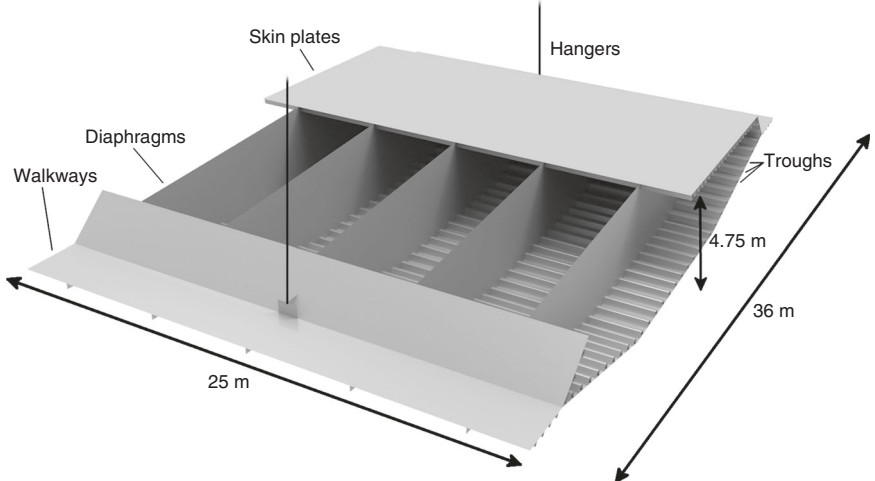

**Fig. 2 Design concept and dimensions of the conventional girder.** A single 25-m-long section of the continuous girder from Fig. 1 is shown with half of the top part removed to reveal the internal details of the solid diaphragms.

engineering is partly due to a conservative industry, owing to often high-cost/high-risk projects, and partly due to the many challenges in structural optimization[25], such as including cost[26] and reducing complexity[27]. Another obstacle is the low volume fractions typically encountered in civil engineering structures (often in the order of a few percent) requiring very fine design resolutions. Finally, it has been a challenge to cover multiple length scales. Bridge girders with spans of kilometers and widths in the order of 20–40 m are built from plates with thicknesses as low as 6 mm. Hence, to discretize the entire periodic girder domain (Supplementary Fig. 1) into finite elements (voxels) with dimensions in the order of the plate thicknesses, and allowing designs with very low volume fractions, requires billions of elements and thus giga-voxel resolution procedures. Until recently, computational morphology methods were limited to a few million elements, however, this limitation was overcome when the methodology was applied to the design of a full-scale Boeing 777 type aircraft wing using 1.1 billion voxels[13].

Here, the giga-scale morphogenesis procedure is further extended and applied to a model of a single 25-m-long girder section, with outer geometry and dimensions as in Fig. 2 (see "Methods" for details). Due to periodicity, a single periodic section is representative, however, to precisely impose boundary conditions and transfer loads, two adjacent sections are included in the three-dimensional continuum model (Supplementary Fig. 1). The focus of the study is the load-carrying internal structure of the main domain; hence, the outer wind profile is maintained while walkways are neglected. The dimensions of the full model are 75 m × 30.1 m × 4.75 m, with a design domain of 12.5 m × 15.05 m × 4.75 m (a quarter of one section). During the optimization process, the design domain is mapped to the rest of the model, ensuring that symmetry is maintained along the two center section axes (Supplementary Fig. 1). The top layer of elements, representing the road surface, and the hanger anchorages are fixed to be solid, whereas material can freely be distributed in the remainder of the domain (Supplementary Fig. 1). Through extensive numerical studies, involving global dynamic and aero-structural modeling of the entire bridge, five representative static load cases are chosen for the optimization procedure. These, including the geometry and dimensions, are all based on the actual design of Turkey's 2682-m-long Osman Gazi Bridge, opened in July 2016 (Fig. 1).

The objective of the optimization is to maximize the stiffness of the center section for a given amount of material. The full model

is discretized into 2.1 billion elements (corresponding to a mesh of 4384 × 1760 × 272 elements) with a maximum element size of 17 mm. Although still above the desired minimum member size of 6 mm, this resolution is found to be sufficient to extract the design trends. If a coarser design resolution had been utilized, identification of the important details of the novel design would not have been possible. The giga-voxel resolution comes at a high cost, however, meaning that it requires access to massive computational resources with run times of up to 85 h on 16,000 CPU cores.

Three sections of the optimized bridge girder design are shown in Fig. 3, with the top layer removed to reveal the resulting internal structures. Apart from the thin fixed top layer and fixed hanger anchorage, no a priori assumptions were made on the geometry. Clearly, the design is very different from the conventional, as no perpendicular diaphragms or orthotropic skin plates are seen. Instead, a number of double-curved diaphragm-like panels and trusses have appeared as result of the morphogenesis process. These unconventional features transfer traffic loads and self-weight directly to the hangers instead of through the inefficient conventional zig–zag patterns. Additionally, a longitudinal plate-like structure appears in the region of the hangers and the curved diaphragms as added support. Furthermore, a decreased span length of the skin plates is possible due to an increased number of supports growing from the diaphragm-like panels. However, the complexity of the optimized design prohibits a direct application in the construction industry due to cost and manufacturing considerations.

**Interpretation of optimized design.** A simplified girder model may be derived by interpreting the main structural features of the optimized design. To facilitate meaningful comparisons to the conventional design, and to convince structural engineers about feasibility, we strive for a derived model at similar level of geometric and manufacturing complexity. Figure 1 shows the best interpreted design together with the conventional layout, and in the lower left-hand side of Fig. 3, the interpreted design is shown overlaid on the optimized design. For simplicity, only diaphragms built from solid plates are considered, however, the principles are equally valid for truss diaphragms. From the rendering of the interpreted design, it is seen that the number of diaphragms per section is increased from five to six, that four of the six diaphragms are curved towards the hangers, and that longitudinal

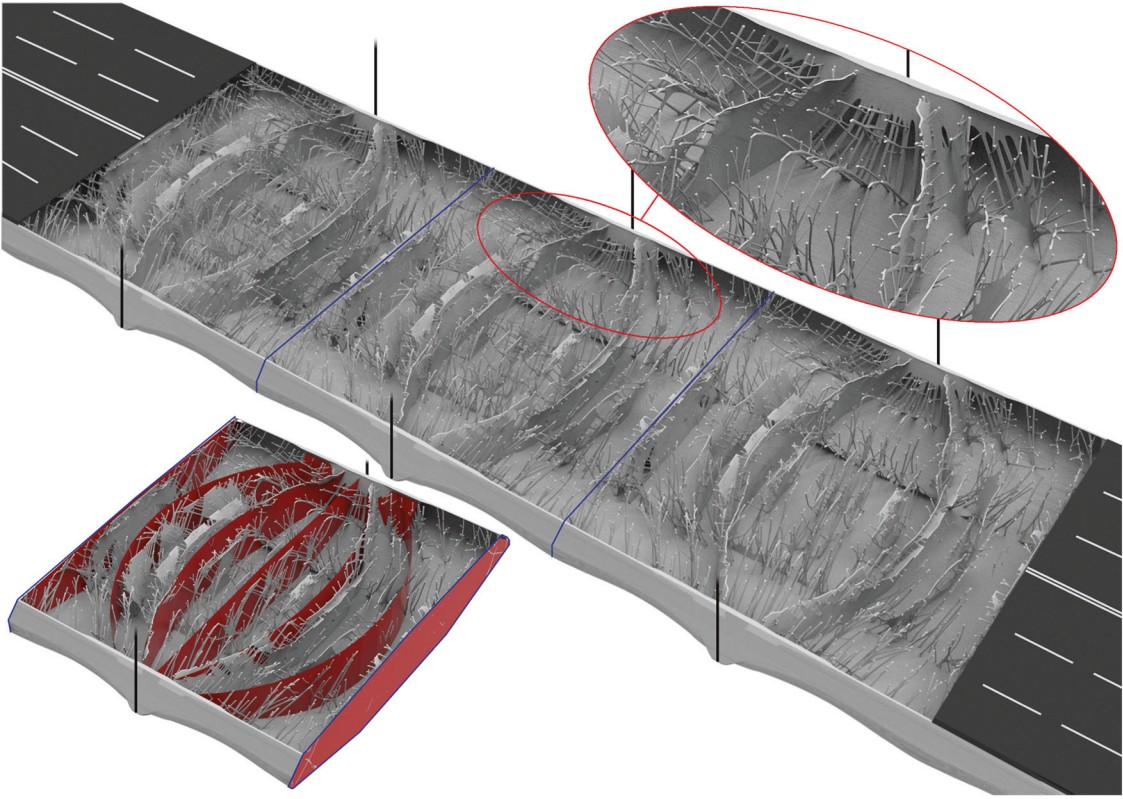

**Fig. 3 Optimized girder structure.** The result of the giga-scale morphogenesis procedure applied to the bridge girder model is shown after 400 steps of optimization using 2.1 billion design variables. Three sections of the continuous girder with the fixed top layer removed reveal the internal details. A single section is shown in the lower left-hand side with overlaid interpreted curved diaphragms (red panels).

panels to support the connection between hanger anchorage and diaphragms have been added. Although the changes appear limited, it is important to recognize that even these seemingly small topological modifications could not have been obtained by classical shape optimization without prior knowledge of the morphogenesis outcome. This confirms the need for the giga-scale resolution free-form design methodology. The interpreted model is subsequently imported and analyzed using a commercial finite element product, revealing a significant 12.7% weight saving compared to the conventional design—just based on a quick interpretation (see Methods for the analysis behind these predictions and Supplementary Table 1).

**Parametric optimization**. From a subsequent sizing optimization of the interpreted design, where the design variables are the thicknesses of the plates (see Fig. 1 and Supplementary Fig. 2), the total weight saving reaches 28.4% (see "Methods" for details). In comparison, a weight saving of 13.8% was achieved in ref. [12], where a similar, but more refined parametric sizing optimization was carried out on the conventional design. Owing to the detail and complexity of the parametric model in ref. [12], the reported 13.8% change provides an upper bound for the conventional design. Hence, the application of the giga-scale morphogenesis procedure, followed by a quick interpretation and a simple parametric optimization, leads to weight savings significantly larger than what was possible by modifying the conventional design through simple parametric modifications.

## Discussion
Although neither the giga-scale optimization nor the parametric model included fatigue in the optimization formulations, we remark that the variation between the largest stresses in the new design is insignificant and absolute values are not bigger than in the conventional design; hence, fatigue problems are not worse than in the conventional design (see Supplementary Table 1).

The interpreted design is preliminary; hence, follow-up studies, including all load cases, dynamics, fatigue, buckling, etc. must be carried out. However, time and cost of these additional studies are trivial when weight becomes the decisive factor. Particularly dynamics, covering e.g. wind vibrations, is of interest due to the significant reduction in girder self-weight. However, we believe that such a detailed redesign process will not change the overall conclusions and insight gained from the presented study. Also, the curved diaphragms leave uneven spans of skin plates, which must be handled, e.g., by redistributing trough and skin plate material from decreased to increased spans. Furthermore, the constructability of the interpreted design must be considered, e.g., by studying the effect of curved diaphragms on construction costs. Despite the slightly more complex design, due to the curved diaphragms, the changes are considered moderate and are repeated between sections. With modern digital production tools and considering recent developments in comparable parametric high-riser designs, we do not believe that these points will be game changers.

The achieved 28.4% weight savings in the girder, equivalent to 8200 tonnes of steel, generate total savings of 13,000 tonnes of steel and 19,000 m³ of concrete in the entire bridge due to knock-on effects (see "Methods" for the analysis behind these estimations and Supplementary Table 2). These savings translate into a reduction of 43,000 tonnes of $CO_2$ emissions, corresponding to 358 million km of car-driving equivalent to 9000 times around the globe. The overall impact of the rather simple adaptions to the conventional design, identified from the results of the giga-scale

morphogenesis procedure, is thus significant and may help to close the gap towards the construction and environmental impact of future's super-long suspension bridges. The achieved savings in weight and $CO_2$ emissions indicate a large potential in applying the demonstrated methods to other civil engineering structures.

## Methods

**General**. First, the finite element model of the bridge girder, subject to optimization, is described, and subsequently, the details of the applied topology optimization methods are introduced. Finally, the methods of interpretation, parametric optimization, and weight-saving estimations are described.

**Bridge girder model**. The optimization studies are based on Turkey's 2682-m-long Osman Gazi Bridge (Fig. 1), which at the opening in July 2016 possessed the World's fourth-longest main span of 1550 m. The COWI-made bridge design, including the orthotropic closed steel box-girder, is considered state-of-the-art, and hence, a suitable basis for optimization and identification of new and innovative bridge girder designs. Geometry, dimensions, and loads are in the following, based on this bridge design.

As a bridge girder is a repetitive periodic structure, a single section, defined by the span between two adjacent sets of hangers (25 m), is representative. However, to impose boundary conditions and global loads, three sections (75 m × 30.1 m × 4.75 m) of the bridge girder are modeled. Hence, a three-dimensional continuum finite element model is established as the design domain, see Supplementary Fig. 1. The domain does not include walkways but maintains the outer shapes defining the aerodynamic wind profile, as aerodynamic optimization is not included in the present work. However, aerodynamic loads are taken into account through the applied loads discussed below. The thin top layer of elements, representing the road surface, and regions near the hanger anchorages, are fixed to be solid (as indicated with green in Supplementary Fig. 1). The entire end surface at one end of the model is fixed, and at the opposite end, global sections forces $N_x$, $M_y$, $M_z$, $V_y$, $V_z$, and $M_t$ are applied, as shown in Supplementary Fig. 1. The global section forces are applied to a stiff end surface, with 100 times the usual modulus of elasticity, to ensure a smooth load transmission into the model. Additionally, local loads are applied in the form of a uniformly distributed load $p$ applied to the fixed solid top surface, and in the form of hanger forces $P$ applied to the fixed hanger anchorages. From a global beam model of the Osman Gazi Bridge, developed during the design work by COWI, a total of 12 sets of critical global load cases were found, corresponding to the maximum and minimum of the six section forces in a beam. The global load cases were extracted from an envelope of load combinations taking account of permanent, traffic, wind, and temperature loads as well as seismic actions. The load combinations and loads were all based on the Eurocode norm and UK National Annexes. Each global load case consists of the static section forces acting at the end of the local continuum model, as well as the local uniformly distributed load on the top surface and hanger forces. Additionally, two local load cases are included, consisting of only hanger forces and equilibrating uniformly distributed load on respectively the entire top surface and half of the top surface (on one side of the longitudinal center axis), hence no global section forces. The 14 load cases are summarized in Supplementary Table 3. We remark that in the presence of varying axial forces, e.g. arising from novel pylon and cable layouts[24], each girder section would have to be designed for a specific loading scenario. This would on one hand allow the overall weight to be reduced even further, but on the other result in girder layouts that are not periodic from section to section. As the main aim of this work is to obtain a single, simplified girder layout, such a study is left for future work.

Through initial studies, the 14 load cases were reduced to the five most important (most influential on structural features during optimization). Hence, only load cases 1, 5, 10, 13, and 14 were included during the final optimization study. Finally, since a bridge girder is symmetric about the longitudinal center axis and transversely symmetric about every hanger set, a symmetry condition is imposed, as indicated in Supplementary Fig. 1. Hence, a quarter of the center section (12.5 m × 15.05 m × 4.75 m) is defined as the active design domain and mapped continuously to the rest of the model during the iterative optimization. Displacement symmetry conditions are not applied, as a measure of reducing the model size, due to asymmetric loading conditions. The material parameters, representing steel, are given as the modulus of elasticity $E_{\text{solid}}$=210 GPa and Poisson's ratio $v = 0.3$.

**Topology optimization**. The objective of the optimization is to maximize the stiffness of the center section of the model for a given amount of material, which during optimization is distributed in the interior of the design domain (marked with orange in Supplementary Fig. 1). The problem is solved based on the assumption of static, linear-elastic behavior. The governing partial differential equations (PDEs) are solved using the finite element method, using eight-node hexahedral isoparametric elements, resulting in the following linear system of equations:

$$\mathbf{K}(\rho)\mathbf{u} = \mathbf{F}. \qquad (1)$$

Here $\mathbf{K}(\rho)$ is the stiffness matrix given as a function of the design variable vector $\boldsymbol{\rho}$,

$\mathbf{u}$ is the displacement vector, and $\mathbf{F}$ is the load vector. Each mesh element is assigned a single artificial continuous density, $\rho_e \in [0, 1]$, collected in vector $\boldsymbol{\rho}$, defined in a range from void (0) to solid material (1). The density-based topology optimization method SIMP[16] (Solid Isotropic Material with Penalization) is used, as this enables the use of gradient-based optimization algorithms. The element stiffness is defined by a smooth interpolation between the solid material stiffness, $E_{\text{solid}}$, and a very weak material ($E_{\text{void}} = 10^{-6}E_{\text{solid}}$), given by the modified SIMP scheme[28] as

$$E(\rho_e) = E_{\text{void}} + \rho_e^p(E_{\text{solid}} - E_{\text{void}}). \qquad (2)$$

Here $p > 1$ is a penalization parameter enforcing almost discrete final designs with $\rho_e$ being either 0 or 1, hence, the impractical intermediate densities are penalized and made unfavorable. The stiffness matrix can thus be assembled in the standard way as

$$\mathbf{K}(\rho) = \sum_e^n E(\rho_e)\mathbf{k}_0^e. \qquad (3)$$

Here $\mathbf{k}_0^e$ is the element stiffness matrix with unit modulus of elasticity and $n$ is the number of elements in the mesh. Before stating the mathematical optimization problem, the objective function is introduced as compliance (work done by external forces) of only the center section of the model, $C = \mathbf{u}^T\mathbf{F}$, which is inversely proportional to structural stiffness. Hence, to maximize structural stiffness, the following minimization problem is posed:

$$\begin{aligned} \min_{\rho \in \mathbb{R}^n} \quad & \phi = \sum_i^L \mathbf{u}_i^T\mathbf{F}_i\alpha_i \\ s.t. \quad & \mathbf{K}(\rho)\mathbf{u}_i = \mathbf{F}_i, \ i = 1\dots L \\ & \frac{V(\boldsymbol{\rho})}{V^*} - 1 \le 0 \\ & 0 \le \rho_e \le 1, \ e = 1\dots n \end{aligned} \qquad (4)$$

Here $\phi$ is the sum of compliances from $L$ load cases, weighted by the scaling factors $a_i$. The first constraint ensures mechanical equilibrium; the second poses a limit on the amount of available material, with $V(\boldsymbol{\rho})$ being the volume of the current structure and $V^*$ the maximum amount of available material and the third defines box constraints on the design variables. To avoid well known, but undesirable effects inherent in topology optimization, such as checkerboards and mesh-dependencies[16], a filtering[29] of the densities is performed after each optimization iteration. Here, the density filter from ref. [13], an image processing type convolution filter, is applied. Hence, the filter modifies each density as a weighted average of the adjacent densities, given by a filter radius. The filter is thus a method to smoothen the boundaries of the structure. A filter radius of 1.5 times the maximum dimension of a mesh-element is used. Finally, a fully parallelized version of the gradient-based method of moving asymptotes (MMA[14,30]) is used as the optimization algorithm.

**Sensitivity analysis**. Since the objective function in problem (4) is the compliance of only the center section of the girder, the sensitivity analysis becomes somewhat more complex since the problem is no longer self-adjoint. The objective function without the scaling factor $a_i$ is given as

$$\phi = \sum_i^L \mathbf{u}_i^T\mathbf{F}_i = \sum_i^L \mathbf{u}_i^T\bar{\mathbf{K}}\mathbf{u}_i. \qquad (5)$$

Here $\bar{\mathbf{K}}$ is a stiffness matrix only containing contributions from the center section of the model. With this formulation, only the compliance of this section is minimized. Undesirable effects from the adjacent sections (including effects from the boundary conditions and stiff end surface) are thus avoided. To ensure the periodic structure, the center domain is mapped to the adjacent sections, as discussed previously. However, when introducing $\bar{\mathbf{K}}$, the optimization problem is no longer self-adjoint, and to find the sensitivities, an adjoint problem must be solved. The sensitivities of Eq. (5) differentiated with respect to the design variable $\rho_e$ are given by

$$\frac{\partial \phi}{\partial \rho_e} = \sum_i^L \mathbf{u}_i^T\frac{\partial \bar{\mathbf{K}}}{\partial \rho_e}\mathbf{u}_i + \boldsymbol{\lambda}_i^T\frac{\partial \mathbf{K}}{\partial \rho_e}\mathbf{u}_i. \qquad (6)$$

Here $\boldsymbol{\lambda}_i$ denote the solutions to the adjoint problems

$$\mathbf{K}^T\boldsymbol{\lambda}_i = -2\bar{\mathbf{K}}\mathbf{u}_i. \qquad (7)$$

Finally, the derivatives of the two stiffness matrices are given as

$$\begin{aligned} \frac{\partial \mathbf{K}}{\partial \rho_e} &= \sum_e^n p\rho_e^{p-1}(E_{\text{solid}} - E_{\text{void}})\mathbf{k}_0^e, \\ \frac{\partial \bar{\mathbf{K}}}{\partial \rho_e} &= \sum_e^{\bar{n}} p\rho_e^{p-1}(E_{\text{solid}} - E_{\text{void}})\mathbf{k}_0^e. \end{aligned} \qquad (8)$$

Here $\bar{n}$ is the number of elements in the center section, equivalent to the contributions to $\bar{\mathbf{K}}$.

**Giga-scale procedure**. To solve the established optimization problem of the described bridge girder model, giga-scale procedures are a necessity. Due to the multiple length scales from the huge domain size of 75 m down to plate thickness as small as 6 mm (in the conventional design), billions of finite elements are required to discretize the domain into a sufficient fineness. The giga-scale morphogenesis procedures presented in ref. [13], based on the PETSc[31–33] framework suitable for large-scale parallel computing, are thus extended to handle the given model and optimization problem. Hence, the domain enclosed by the outer dimensions in Supplementary Fig. 1 is discretized into 2.1 billion elements (corresponding to a mesh of 4384 × 1760 × 272 elements) with a maximum element size of 17 mm. Despite the element size being about three times larger than the smallest plate thickness in the conventional design, the fineness of the mesh is capable of revealing intriguing and detailed new structural features. The allowable volume fraction is chosen as $V^* = 3.0\%$, close to the typical volume fractions of 1.0–1.5% in the conventional design (1.3% in the Osman Gazi Bridge). The slightly larger volume fraction is chosen to ensure detailed results, in connection with the corresponding larger element size in the model. The purpose of the optimization should meanwhile be clear to the reader: to give qualitative insight into how an optimized girder design could look, and thus not to show a final optimal structure. An exact volume fraction is thus not crucial for the study in question.

Considering the quite simple final result of our study, i.e. exchange original five straight spars with six curved ones, one could rightfully object that an expensive supercomputing procedure, as suggested, is overkill and a simple shape optimization procedure could have been sufficient. However, this outcome could not have been predicted without it. A priori, it was not known that this diaphragm layout and placement was key to the weight reduction. A change from five to six diaphragms would require more than a simple shape optimization process, and the bending of each of the diaphragms would have required knowledge of the required parameterization.

**Ensuring structural details**. Since the optimization problem (4) is non-convex, due to the stiffness-penalization, any gradient-based solution method will end up in a local minimum. To ensure high-quality designs (strong local minima), and to allow for a smooth and fast convergence, a continuation strategy is applied for the penalization parameter in the SIMP interpolation. In the strategy, similar to the one in ref. [13], the penalization parameter is slowly and smoothly raised in steps of 0.25 from 1 to 3 over a total of 400 design cycles.

**Computations**. To solve the giga-scale optimization problem requires a massive amount of computational resources. The presented results were generated on the Joliet-Curie Supercomputer in France. The optimal number of processors capable of solving the problem was found by numerical studies and memory requirements to be around 16,000. Notably, the symmetry mapping require heavy memory use. During each of the 400 design cycles, ten PDEs (five load cases and five adjoint problems) with around 6.3 billion degrees of freedom are solved with an average solver time of 35 s per equation. The total run time, including IO and check-pointing for restarts, was 85 h.

**Interpretation**. Since the results of the morphogenesis procedure are mainly qualitative, a quantitative study is performed to determine an estimated figure of weight savings. Hence, based on the insight from the main structural features of the results in Fig. 3, the interior of the conventional design is adapted into an interpreted design. This is identified by visual examination of the results in Fig. 3, and found as the best among a screening of multiple initial interpretations. For simplicity, only diaphragms of solid plates are considered. Here, the number of diaphragms per section is increased from five to six, four of the six diaphragms are curved towards the hangers, and longitudinal panels to support the connection between hanger anchorage and diaphragms are added (called "hanger steel plates"), see Fig. 1. Subsequently, both the conventional design (Figs. 1 and 2) and the interpreted design (Fig. 1) are modeled with shell elements in the commercial finite element software Abaqus (by Simulia™). The models include skin plates with troughs, walkway plates with troughs, and transverse diaphragm plate panels. The models have identical material volumes, and the plate thickness of the different parts are summarized in Supplementary Table 4 (see Supplementary Fig. 2 for designation). The shell models represent the three girder sections previously described with similar support conditions, loads, and material parameters.

To quantify the performance improvement, the weighted compliance of the center section, Eq. (5), is computed for the five representative load cases used in the initial topology optimization. The numerical results are summarized in Supplementary Table 1. In column two, the weighted compliance of the five load cases is seen. In column three, the improvement is shown, hence the interpreted design is seen to be 12.7% stiffer than the conventional for the five load cases. The change in compliance, or stiffness, can generally be translated to an equivalent change in volume, or weight. Hence, the stiffness increase may be seen as a weight reduction.

To ensure that the five chosen load cases are representative for the original 14, the weighted compliance of all 14 load cases is computed as well for the shell models. The results are seen in column four of the table. Since the improvement,

seen in column five, is similar to the figures in column three, the five load cases are concluded indeed to be representative. Furthermore, the average of the maximum von Mises stress in each of the five load cases is included in column six, with the change shown in column seven. Since the change in stresses is small, fatigue problems are not considered to increase in the improved design.

**Parametric optimization**. In addition to the interpretation and quantitative comparison above, additional parametric optimization is carried out on the interpreted design. The Abaqus shell model of the interpreted design is used for the parametric optimization, which is carried out in the commercial software iSight (by Simulia™). The objective of the optimization is to minimize the compliance in the center section of the model without increasing the amount of material, hence similar to the topology optimization problem, Eq. (4). However, the design variables are now the plate thickness of the shell elements, as indicated in Supplementary Fig. 2 and Supplementary Table 4. A lower bound of 4 mm plate thickness is imposed, similar to the studies in ref. [12]. The non-linear optimization problem is solved by sequential quadratic programming with the use of finite difference gradients.

The results of the parametric optimization are summarized in Supplementary Table 1 and Supplementary Table 4. From the first of the two tables, it is seen that the parametric optimization leads to a total weight saving of 28.4% compared to the conventional design. Besides, when studying the weighted compliance of the 14 load cases and the von Mises stresses, of the design after parametric optimization, similar conclusions as above are reached.

In Supplementary Table 4, it is seen that all troughs reach the lower limit of 4 mm, similar to the trend seen in ref. [12]. Furthermore, it is seen that the thickness of the inclined web plate, located next to the hangers, increases significantly. However, this can be explained from the simplicity of the shell model, since the, in reality, structural complex high-stress region near the hangers, is modeled with few plates. Hence, during optimization, the thickness of the inclined web plate increases to reduce the compliance near the hangers.

**Estimation of knock-on effects and CO₂ emission savings**. Due to the load-carrying principles of suspension bridges, all weight savings in the bridge girder move up the load path. The potential knock-on effects are here estimated based on conservative, but fair assumptions. The material quantities of the Osman Gazi Bridge are 33,600 tonnes of girder steel (with walkway), 18,000 tonnes of main cable steel, 17,000 tonnes of tower steel, 45,000 m³ of tower foundation concrete, and 130,000 m³ of anchor block concrete.

The 28.4% (8,200 tonnes) weight saving in structural steel in the girder (without walkways), translates into a reduction in loads transferred to the cables of 19.1% when adjusting for the contribution from walkways (about 11% of self-weight) and 60 mm surfacing (about 22% of self-weight). Conservatively, the weight saving in the main cables is equivalent to the 19.1% reduction, when minor effects from cable self-weight are disregarded. Further, the main cable saving translates directly to the anchor block saving. However, in the towers, the saving is halved to 9.55% since the normal force (from the main cable) only contributes with roughly 50% of all tower forces (with the remaining 30% from bending moments and 20% from buckling). Finally, the saving in the tower foundations is equivalent to 9.55%. From the above-estimated knock-on savings, the total steel and concrete quantities can be found, which are summarized in Supplementary Table 2, along with the savings in the interpreted design before and after parametric optimization.

To estimate the CO₂ emissions, the following values (from the ICE Inventory of carbon and energy v3.0 Database) are used; 2460 kgCO₂e per ton for steel plates (with density 7850 kg per m³) and 150 kgCO₂e per ton for concrete (with density 2400 kg per m³). The emissions are only calculated based on the material quantities, hence disregarding construction methods. The results are summarized in Supplementary Table 2.

## Data availability

The results and data sets presented in this work, i.e. the optimized designs in the form of STL files as well the Abaqus models including loads, are freely available online from https://doi.org/10.11583/DTU.c.4962806.

## Code availability

The basic C++/MPI code used for the giga-scale optimization is publicly available at https://github.com/topopt/TopOpt_in_PETSc (ref. [14]).

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

## Acknowledgements
The presented work is part of an industrial ph.d. project with the title "Innovative design of steel girders in cable-supported bridges" and is carried out in cooperation with COWI A/S, DTU Civil Engineering, and DTU Mechanical Engineering. The project is funded by the COWI Foundation grant C-131.02 and Innovation Fund Denmark grant 5189-00112B. This work was further funded through a PRACE (Partnership for Advanced Computing in Europe) grant TopBridge giving access to the Joliet-Curie supercomputer. Access to and efficient support from Myriam Peyrounette at IDRIS.fr is highly appreciated. The authors acknowledge the support of the Villum Foundation through the Villum Investigator Project InnoTop.

## Author contributions
M.B. contributed to the method development, implementation, interpretation, visualization, and manuscript preparation. O.S. contributed to the original idea, method development, interpretation, and manuscript preparation. H.P. contributed to the original idea, method development, and manuscript preparation. N.A. contributed to the method development, implementation, interpretation, supercomputing, visualization, and manuscript preparation.

## Competing interests
The authors declare no competing interests.
