## [Peer Review File · Nature Communications]

Reviewers' Comments:

Reviewer #1:

Remarks to the Author:

The paper is concerned with reducing the weight of the girder carrying the traffic loading in super-long span suspension bridges. Self-weight becomes a constraint limiting the achievable span of such bridges so identifying more materially efficient girder designs is certainly valuable. As noted in the paper, reducing deck weight also reduces the amount of material required in the main cables and pylons.

The authors cite CO2 reductions, and in large structures such as super-long span suspension bridges the reductions are large; on the other hand, given the relatively small number of long span bridges constructed each year, the % impact of the development described on global emissions arising from construction globally will likely be comparatively small.

The "computational morphogenesis", or topology optimization, procedure employed is not novel, rather the application and the high resolution of the model employed is novel.

As the authors indicate, previous use of topology optimization in the civil engineering domain has been relatively limited. This is in part due to the low volume fractions typically encountered in civil engineering problems (often of the order of 1% or less). This is currently covered only indirectly in the text; it would be useful if the authors could mention this more explicitly, as it helps justify the use of the high numerical resolution employed.

Although the topology optimization output is not used directly, it is used to infer a geometrically simple structural model - conceptually similar to the approach used in [24] when seeking efficient arrangements of cables and pylons in long span bridges. (In this latter paper suspension bridge forms are identified as being less efficient than proposed new forms of cable supported bridge structures, albeit under idealized conditions. The proposed topology optimization inspired design approach for girders could potentially be used in combination with such these new forms to maximize overall material utilization, albeit the girder designs would need to vary across the span due to the presence of varying axial forces.)

Although considering the huge CPU resource required the reduction in material usage achieved could be viewed as quite modest (28.4% vs 13.8% when performing parametric optimization alone), my view is that the contribution is important and warrants publication. The paper is well written and is suitably illustrated.

Other suggestions:

- In Fig.3 it would be useful to identify cable support points more clearly.
- It is stated that the curved diaphragms leave uneven spans of skin plates, "which must be handled"; it would probably be helpful to add a sentence to clarify how this is achieved.
- Line 307: change "the sensitivity analysis complicates slightly" to "the sensitivity analysis becomes more complex".
- If practicable, it would be useful if the full topology optimization output and simplified model could be made available to the community; this could e.g. lead to alternative simplified designs being identified.

Reviewer #2:

Remarks to the Author:

Review of: Closing the gap towards super-long suspension bridges

For: Nature Communications

This paper presents a large-scale freeform design of the interior of a bridge girder, reducing the weight sufficiently to open up for design of super-long suspension bridges. The Osman Gazi Bridge with 1.5 km free span is used as case study. It thus presents novel use of high-resolution freeform design for bridges. The topic of reducing the weight (and thus carbon emissions) of the built civil environment is, as the authors state, of high relevance. The technical content of the paper is therefore recommended for publication after major revisions.

The paper does not always read well. Some sentences are very long and some key aspects of the work are not communicated clearly. This should be addressed prior to publication.

Major comments:

- It is unclear to the reviewer why it is needed to do such a large scale free-form design (84 hours on a super computer), if the end goal is to place six plate diaphragms. The authors should make clear why this enormous computational effort was chosen as opposed to conducting shape optimization with a couple of different starting guesses.
- The constructability of the new design is not discussed. The authors mention that constructability has been an important factor preventing freeform design of civil structures. The new design curves and has connections that are angled differently. What does that mean for labor costs, outcome quality, etc.?
- The authors state that construction is accountable for ~40% of the annual carbon emissions and use that to motivate their low weight design of a super long bridge. One could argue, that although these bridges use a lot material, there are not globally enough being built yearly to make a real indent on the enormous emissions from construction. How would the authors respond to that?
- The section on the sensitivity analysis is confusing. Consider writing the objective as in Eq. (4).
- Line 356: How was the interpretation done? Please provide details on how the six diaphragm plates were identified.
- Please provide details on the load cases. Are they all static?
- The paper would benefit from adding a section on wind vibrations. Intuitively, a lighter bridge would be more prone to severe vibrations. How does the response compare for the two designs in Fig. 1?

Reviewer #1 (Remarks to the Author):

The paper is concerned with reducing the weight of the girder carrying the traffic loading in super-long span suspension bridges. Self-weight becomes a constraint limiting the achievable span of such bridges so identifying more materially efficient girder designs is certainly valuable. As noted in the paper, reducing deck weight also reduces the amount of material required in the main cables and pylons.

The authors cite CO₂ reductions, and in large structures such as super-long span suspension bridges, the reductions are large; on the other hand, given the relatively small number of long span bridges constructed each year, the % impact of the development described on global emissions arising from construction globally will likely be comparatively small.

Indeed, the number of long span bridges constructed each year is relatively small; hence, the % impact on construction globally will be small if only considering this specific application. However, we see our bridge example as an eye-opener since the methods and outcomes described in the manuscript are generally applicable to civil engineering. The strategy of first applying a giga-scale morphogenesis procedure and then subsequently making an interpretation to simpler and constructible designs, is generally applicable. In the manuscript, the methods were demonstrated on a specific structure, girders in super-long suspension bridges, to illustrate the potential (see existing lines 67-70 in the revised manuscript). It is our hope that our bridge example will open the eyes of the civil engineering industry with regards to similar saving potentials in other applications with significant expected CO₂ reductions as a result. To make this aspect clearer we have modified and extended the manuscript in lines 70-71 and 169-171.

The "computational morphogenesis", or topology optimization, procedure employed is not novel, rather the application and the high resolution of the model employed is novel.

We completely agree. For the same reason we have not made the specifics of the used topology optimization approach the key message of the paper. The main message is to use the giga-scale approach to reach relevant length-scales/volume fractions – and then to interpret on the result, which results in the simple curved diaphragm design. Hence, the contribution is the result and its potential for changing civil engineering. We remark, however, that the extension of our computational morphogenesis approach to girder design was not trivial, partly because of even more degrees of freedom, but more so due to the complexity of invoking symmetry conditions in the super large scale parallel computation framework.

As the authors indicate, previous use of topology optimization in the civil engineering domain has been relatively limited. This is in part due to the low volume fractions typically encountered in civil engineering problems (often of the order of 1% or less). This is currently covered only indirectly in the text; it would be useful if the authors could mention this more explicitly, as it helps justify the use of the high numerical resolution employed.

We completely agree on this point. We have elaborated on this observation explicitly in lines 87-89 and 91-93 of the revised manuscript.

Although the topology optimization output is not used directly, it is used to infer a geometrically simple structural model - conceptually similar to the approach used in [24] when seeking efficient arrangements of cables and pylons in long span bridges. (In this latter paper suspension bridge forms are identified as being less efficient than proposed new forms of cable supported bridge structures, albeit under idealized conditions. The proposed topology optimization inspired design approach for girders could potentially be used in combination with such these new forms to maximize overall material utilization, albeit the girder designs would need to vary across the span due to the presence of varying axial forces.)

This is indeed an interesting idea. As indicated, this would require a slightly different approach due to the varying axial force, resulting in girders that are not periodic from section to section. Hence, other sectional forces will be governing and each section (or key sections) should undergo the design approach. On the other hand, we note that such an approach would require custom shapes for each section of the bridge – something that may overwhelm an already conservative civil engineering industry. For suspension bridges, the design is known to be very repetitive between the hangers as local design loads are almost constant throughout the entire bridge and global design loads vary only very little throughout the bridge. With the presently imposed periodicity, all sections will be identical, albeit still challenging to manufacture compared to existing girders. We have added a discussion of varying loads on the design problem to the methods sections, see lines 271-275 in the revised manuscript.

Although considering the huge CPU resource required the reduction in material usage achieved could be viewed as quite modest (28.4% vs 13.8% when performing parametric optimization alone), my view is that the contribution is important and warrants publication. The paper is well written and is suitably illustrated.

Other suggestions:

- In Fig.3 it would be useful to identify cable support points more clearly.

Cables have been added in Fig. 3 in the revised manuscript to identify support points more clearly.

- It is stated that the curved diaphragms leave uneven spans of skin plates, "which must be handled"; it would probably be helpful to add a sentence to clarify how this is achieved.

We agree and a comment on how to achieve this has been added in lines 156-158 in the revised manuscript.

- Line 307: change "the sensitivity analysis complicates slightly" to "the sensitivity analysis becomes more complex".

This has been changed, see line 324 in the revised manuscript.

- If practicable, it would be useful if the full topology optimization output and simplified model could be made available to the community; this could e.g. lead to alternative simplified designs being identified.

Both the topology optimization result as well as the simplified model will be made publicly available as STLfiles from a DTU based web-server. And also from Nature Communications if they have the possibility of hosting very large data sets.

Reviewer #2 (Remarks to the Author):

Review of: Closing the gap towards super-long suspension bridges

For: Nature Communications

This paper presents a large-scale freeform design of the interior of a bridge girder, reducing the weight sufficiently to open up for design of super-long suspension bridges. The Osman Gazi Bridge with 1.5 km free span is used as case study. It thus presents novel use of high-resolution freeform design for bridges. The topic of reducing the weight (and thus carbon emissions) of the built civil environment is, as the authors state, of high relevance. The technical content of the paper is therefore recommended for publication after major revisions.

The paper does not always read well. Some sentences are very long and some key aspects of the work are not communicated clearly. This should be addressed prior to publication.

We have carefully reread the manuscript and made several smaller and larger edits to improve readability – specifically with regards to your comments below.

Major comments:

- It is unclear to the reviewer why it is needed to do such a large scale free-form design (84 hours on a super computer), if the end goal is to place six plate diaphragms. The authors should make clear why this enormous computational effort was chosen as opposed to conducting shape optimization with a couple of different starting guesses.

We thank the reviewer for making us aware that the reasoning for the expensive free-form design approach has not been conveyed clearly in the original manuscript. It was definitely not our goal to look for a way to put six plate diaphragms. This was only the (positive and surprisingly simple) outcome of an unrestricted design process that potentially could have resulted in something much more geometrically complex.

We do not believe that this outcome could have been predicted without the preceding free-form approach. Without it, we could not have known that the diaphragm layout and placement was the key to the obtained weight reduction. The change from five to six diaphragms would require more than a simple shape optimization process and the bending of each of the diaphragms would have required one to know the required parameterization a priori. Anyway, if somehow conceived, such a design parameterization and associated remeshing and iterative process, would most probably require weeks of manpower spent with CAD and FEtools. We have added a small discussion of the above to line 357-363 in the revised manuscript.

- The constructability of the new design is not discussed. The authors mention that constructability has been an important factor preventing freeform design of civil structures. The new design curves and has connections that are angled differently. What does that mean for labor costs, outcome quality, etc.?

A brief discussion on constructability of the new design has been added in lines 158-160 in the revised manuscript. Indeed, the new design is more complex but on the other hand, the changes are moderate and equal between sections. With modern digital production tools and considering the development in non-simple high-riser designs, we do not believe that these points will be game changers. As stated in lines 155-156 of the manuscript, however, the interpreted design is preliminary and follow-up studies must be carried out.

- The authors state that construction is accountable for ~40% of the annual carbon emissions and use that to motivate their low weight design of a super long bridge. One could argue, that although these bridges use a lot material, there are not globally enough being built yearly to make a real indent on the enormous emissions from construction. How would the authors respond to that?

A similar point was made by the first reviewer and we repeat the response here:

Indeed, the number of long span bridges constructed each year is relatively small; hence, the % impact on construction globally will be small if only considering this specific application. However, we see our bridge example as an eye-opener since the methods and outcomes described in the manuscript are generally applicable to civil engineering. The strategy of first applying a giga-scale morphogenesis procedure and then subsequently making an interpretation to simpler and constructible designs, is generally applicable. In the manuscript, the methods were demonstrated on a specific structure, girders in super-long suspension bridges, to illustrate the potential (see existing lines 67-70 in the revised manuscript). It is our hope that our bridge example will open the eyes of the civil engineering industry with regards to similar saving potentials in other applications with significant expected CO₂ reductions as a result. To make this aspect clearer we have modified and extended the manuscript in lines 70-71 and 169-171.

- The section on the sensitivity analysis is confusing. Consider writing the objective as in Eq. (4).

We agree. The sensitivity analysis section has been rewritten in the revised manuscript using the objective as stated in Eq. (4).

- Line 356: How was the interpretation done? Please provide details on how the six diaphragm plates were identified.

The interpretation was first done visually based on the structure evolving from the giga-scale topology optimization. However, multiple (a total of 6) interpretations were made and tested initially, before the presented interpreted design was identified as the best. As shown in Fig. 3, the overlaid interpreted design resembles the main structural features of the high-resolution design. The other considered interpretations tested more or less diaphragms and their curvatures as well as placement of reinforcement plates. This procedure of interpretation has been elaborated on in the revised manuscript in lines 381-382.

- Please provide details on the load cases. Are they all static?

Indeed, all load cases are static. The load cases are identified from a global beam model of the Osman Gazi Bridge. This beam model was used during the actual detailed design stage of the bridge and thus consists of envelopes of the most critical load cases taking account of both permanent, traffic, wind, seismic, and temperature loads. The design of the Osman Gazi Bridge, and thus the loads used in the present work, was done according to the Eurocode norms (EN 1991-1-1, EN 1991-2, EN 1991-1-4, EN 1991-1-5) and the UK National Annex.

Since the present study is considered to be on a conceptual level, information on exact load cases and combinations have not been included, as this is not relevant for the work and the results. However, since it was important to include loads with realistic magnitudes, the global model from the detailed design stage was used.

More details on the load cases has been provided in the revised manuscript, please see lines 262-265.

- The paper would benefit from adding a section on wind vibrations. Intuitively, a lighter bridge would be more prone to severe vibrations. How does the response compare for the two designs in Fig. 1?

We agree that dynamical effects would be very interesting to study. This includes both the effect of wind vibrations as well as traffic-induced vibrations when reducing the girder self-weight significantly. As discussed above, such loads were already taken care of in the original load cases, albeit for the higher weight. Surely, recalculating all loads based on a lighter structure and performing a new optimization would yield a different design and hence demanded in practice. However, we believe that such a detailed redesign process will not change the overall conclusions and insight gained from the presented study. We have briefly elaborated on this in connection with the listing of suggested follow-up studies, which includes dynamics, c.f. lines 155-157 in the revised manuscript.

Reviewers' Comments:

Reviewer #2:

Remarks to the Author:

All my comments have been addressed. The authors are recommended to bring more of the discussions from their response to reviewer comments into the paper. Many responses (e.g. regarding constructibility, vibrations and the need for using free form design) are discussed well in the response document, but are only reflected by a sentence or two in the revised manuscript.

The authors would once more like to thank the reviewer for her/his constructive and valuable comments based upon which we have modified and updated the manuscript. Below please find response to your comments. All changes in the revised manuscript are marked in red.

Mads Baandrup

Reviewer #2 (Remarks to the Author):

All my comments have been addressed. The authors are recommend to bring more of the discussions from their response to reviewer comments into the paper. Many responses (e.g. regarding constructibility, vibrations and the need for using free form design) are discussed well in the response document, but are only reflected by a sentence or two in the revised manuscript.

More of the discussions from the previous reviewer response have been included in the revised manuscript. Please see lines 62, 109-112, 140-143, 164-167, and 170-174 in the revised manuscript.